# Human Milk Oligosaccharide Utilization in Intestinal Bifidobacteria Is Governed by Global Transcriptional Regulator NagR

Aleksandr A. Arzamasov,[a] Aruto Nakajima,[b] Mikiyasu Sakanaka,[b] Miriam N. Ojima,[b] Takane Katayama,[b] Dmitry A. Rodionov,[a] Andrei L. Osterman[a]

aInfectious and Inflammatory Diseases Center, Sanford Burnham Prebys Medical Discovery Institute, La Jolla, California, USA
bGraduate School of Biostudies, Kyoto University, Kyoto, Japan

**ABSTRACT** *Bifidobacterium longum* subsp. *infantis* is a prevalent beneficial bacterium that colonizes the human neonatal gut and is uniquely adapted to efficiently use human milk oligosaccharides (HMOs) as a carbon and energy source. Multiple studies have focused on characterizing the elements of HMO utilization machinery in *B. longum* subsp. *infantis*; however, the regulatory mechanisms governing the expression of these catabolic pathways remain poorly understood. A bioinformatic regulon reconstruction approach used in this study implicated NagR, a transcription factor from the ROK family, as a negative global regulator of gene clusters encoding lacto-*N*-biose/galacto-*N*-biose (LNB/GNB), lacto-*N*-tetraose (LNT), and lacto-*N*-neotetraose (LNnT) utilization pathways in *B. longum* subsp. *infantis*. This conjecture was corroborated by transcriptome profiling upon *nagR* genetic inactivation and experimental assessment of binding of recombinant NagR to predicted DNA operators. The latter approach also implicated *N*-acetylglucosamine (GlcNAc), a universal intermediate of LNT and LNnT catabolism, and its phosphorylated derivatives as plausible NagR transcriptional effectors. Reconstruction of NagR regulons in various *Bifidobacterium* lineages revealed multiple potential regulon expansion events, suggesting evolution from a local regulator of GlcNAc catabolism in ancestral bifidobacteria to a global regulator controlling the utilization of mixtures of GlcNAc-containing host glycans in *B. longum* subsp. *infantis* and *Bifidobacterium bifidum*.

**IMPORTANCE** The predominance of bifidobacteria in the gut of breastfed infants is attributed to the ability of these bacteria to metabolize human milk oligosaccharides (HMOs). Thus, individual HMOs such as lacto-*N*-tetraose (LNT) and lacto-*N*-neotetraose (LNnT) are considered promising prebiotics that would stimulate the growth of bifidobacteria and confer multiple health benefits to preterm and malnourished children suffering from impaired (stunted) gut microbiota development. However, the rational selection of HMO-based prebiotics is hampered by the incomplete knowledge of regulatory mechanisms governing HMO utilization in target bifidobacteria. This study describes NagR-mediated transcriptional regulation of LNT and LNnT utilization in *Bifidobacterium longum* subsp. *infantis*. The elucidated regulatory network appears optimally adapted to simultaneous utilization of multiple HMOs, providing a rationale to add HMO mixtures (rather than individual components) to infant formulas. The study also provides insights into the evolutionary trajectories of complex regulatory networks controlling carbohydrate metabolism in bifidobacteria.

**KEYWORDS** bifidobacteria, HMO, regulon, comparative genomics, evolution, prebiotics, transcription factor, carbohydrate metabolism

Address correspondence to Aleksandr A. Arzamasov, arzamasov.alexander@gmail.com, or Andrei L. Osterman, osterman@sbpdiscovery.org.

The authors declare a conflict of interest. A.L.O. and D.A.R. are co-founders of Phenobiome Inc., a company pursuing the development of computational tools for predictive phenotype profiling of microbial communities. Employment of M.S. and M.N.O. at Kyoto University is supported by Morinaga Milk Industry Co., Ltd.

**B**ifidobacteria are Gram-positive, anaerobic, saccharolytic microorganisms that colonize the digestive tracts of humans and various animals (1). Certain *Bifidobacterium* species, namely, *Bifidobacterium longum* subsp. *infantis*, *Bifidobacterium longum* subsp. *longum*, *Bifidobacterium bifidum*, and *Bifidobacterium breve* often predominate the human neonatal

gut microbiota (GM) during breastfeeding (2–5), and their predominance is directly linked with the healthy development of the infant (6, 7). The health-promoting effects attributed to the infant-associated bifidobacteria include protection from enteropathogen colonization (8–10) and modulation of the immune system (11, 12).

Decreased *Bifidobacterium* abundance is characteristic of immature GMs observed in preterm infants (13) and children suffering from severe acute malnutrition (14, 15). Therapeutic approaches aimed at restoring bifidobacterial population in these affected groups include administering exogenous *Bifidobacterium* species (e.g., *B. longum* subsp. *infantis*) as probiotics and/or food formulas containing prebiotics that would selectively stimulate the growth of autochthonous bifidobacteria in the gut and thus confer beneficial properties to the infant (13, 15–17). Since the prevalence of bifidobacteria in the neonatal gut is often attributed to their ability to selectively utilize dietary human milk oligosaccharides (HMOs) (18–20), these milk glycans are considered "natural" prebiotics and added to infant formulas (21).

HMOs are the third most abundant (5 to 20 g/L) component of human milk after lactose (Lac) and lipids and are not assimilated by the infant (22). HMO building blocks include glucose (Glc), galactose (Gal), *N*-acetylglucosamine (GlcNAc), L-fucose (Fuc), and *N*-acetylneuraminic acid (Neu5Ac); these units form more than 100 linear or branched oligosaccharide species (23, 24). Most HMOs contain a Lac core (Gal$\beta$1-4Glc) at the reducing end. The Lac core can be elongated at the C-3 position of the galactose residue with a lacto-*N*-biose (LNB; Gal$\beta$1-3GlcNAc) or *N*-acetyllactosamine (Gal$\beta$1-4GlcNAc) unit(s) (23, 24). The resulting HMO structures are denoted as type I and type II chains, respectively, with lacto-*N*-tetraose (LNT) and lacto-*N*-neotetraose (LNnT) as archetypes. Type I/II chains and the Lac core are often decorated by Fuc and Neu5Ac residues via various $\alpha$-glycosidic bonds.

Previous studies have revealed substantial variation in HMO utilization strategies and capabilities within the *Bifidobacterium* genus (25–29). For example, *B. bifidum* uses a set of membrane-attached glycoside hydrolases (GHs) that extracellularly degrade HMOs to di- and monosaccharides (30–35) and then imports and catabolizes liberated LNB and Lac (25). In contrast, *B. longum* subsp. *infantis* and *B. breve* import HMOs using ATP-binding cassette (ABC) transporters and then degrade the oligosaccharides taken up to monosaccharides intracellularly using a repertoire of exo-acting GHs (36).

*B. longum* subsp. *infantis*, which is widely used in probiotic and synbiotic formulations (8, 15–17, 37), possesses several unique gene clusters (e.g., HMO cluster I or H1) that encode the most elaborate HMO uptake (38–40) and intracellular degradation machinery (41–44) among bifidobacteria (Fig. 1A and B). While the molecular mechanisms of HMO utilization by *B. longum* subsp. *infantis* have received considerable attention, the regulatory mechanisms governing this catabolic process remain poorly understood. Previous studies demonstrated that LNT and LNnT induce a profound and surprisingly similar transcriptomic response in *B. longum* subsp. *infantis* ATCC 15697, with multiple gene clusters (*nag*, *lnp*, H1) (Fig. 1B) being upregulated, pointing to a possible global regulatory mechanism(s) (45, 46). Uncovering this yet unknown mechanism, in addition to fundamental importance, has a potential translational value for the rational selection of individual HMOs for prebiotic and synbiotic formulations.

Comparative genomic analysis across multiple related genomes is a powerful approach for reconstructing transcriptional regulatory networks (regulons) controlling carbohydrate utilization (47–49). Our earlier bioinformatic analysis conducted on a limited set of *Bifidobacterium* genomes implicated NagR, a transcription factor (TF) from the ROK family, as a regulator of *nag* and *lnp* clusters (Fig. 1B) encoding GlcNAc and lacto-*N*-biose/galacto-*N*-biose (LNB/GNB) catabolic pathways, respectively, in *B. bifidum*, *B. breve*, *B. longum* subsp. *infantis*, and *B. longum* subsp. *longum* (48). James et al. further experimentally confirmed this prediction in *B. breve* UCC2003 (50). However, it was unclear whether and how this knowledge translated to the global regulation of the extensive HMO utilization machinery in *B. longum* subsp. *infantis*.

Here, we reconstructed NagR regulons in a substantially larger collection of *Bifidobacteriaceae* genomes focusing on HMO-utilizing species. This analysis revealed multiple putative NagR-binding sites (operators) in the *B. longum* subsp. *infantis* ATCC 15697 genome, suggesting the role of NagR as a global negative regulator of LNB/GNB, LNT, LNnT,

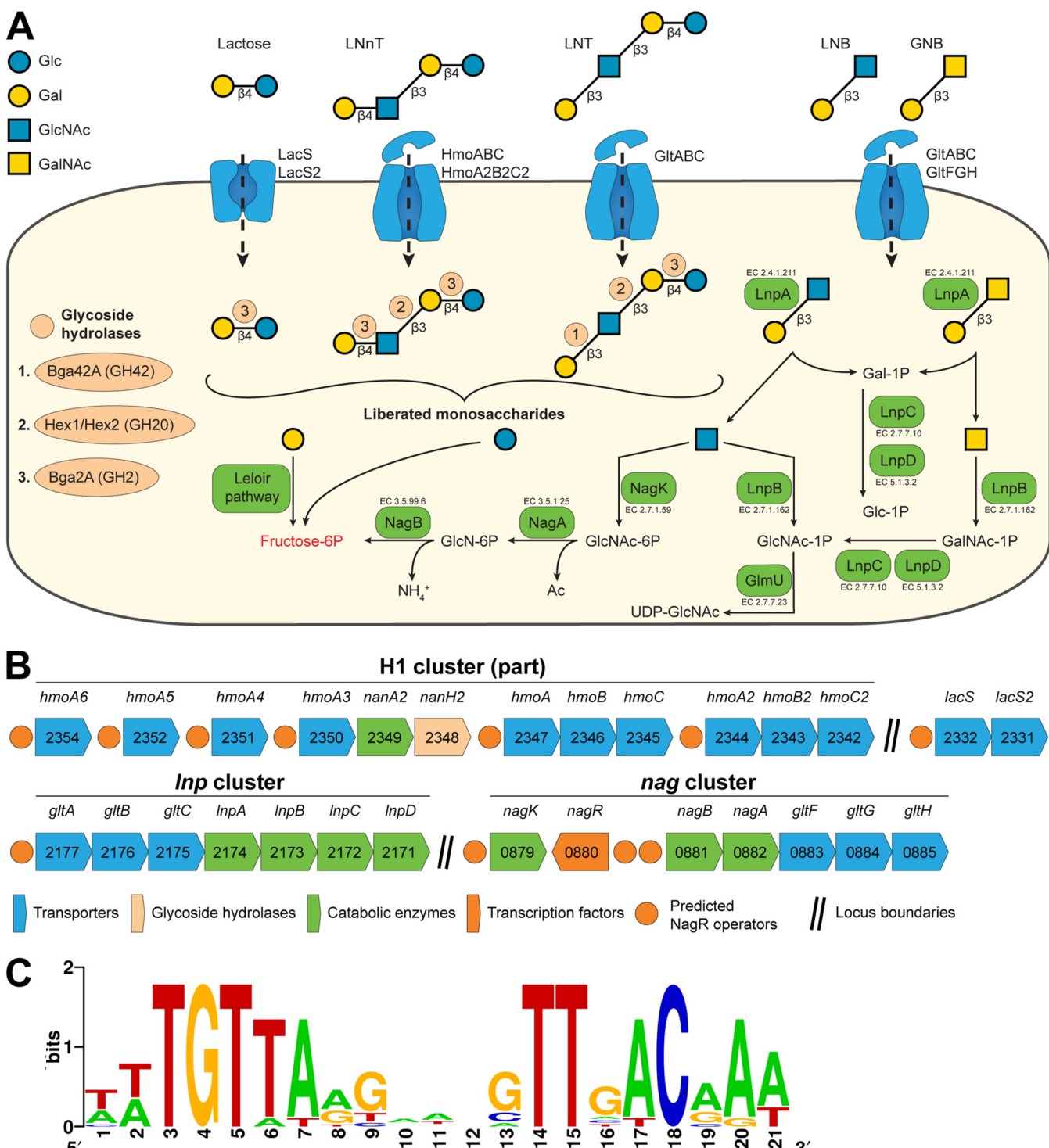

**FIG 1** Reconstructed NagR regulon in *B. longum* subsp. *infantis* ATCC 15697. (A) Schematic representation of LNT, LNnT, and LNB/GNB utilization pathways in *B. longum* subsp. *infantis* ATCC 15697. (Step 1) HMOs and their constituents are transported into the cell by various transport systems. (Step 2) Once inside the cell, HMOs are degraded from the nonreducing end by a coordinated action of exo-acting GHs. Breakdown of glycosidic bonds by specific GHs is indicated by light orange circles. (Step 3) Released monosaccharides are converted to fructose-6P and enter the bifid shunt. (B) Gene clusters constituting the reconstructed NagR regulon in *B. longum* subsp. *infantis* ATCC 15697. Numbers represent locus tags in the Blon_XXXX format (GenBank accession no. CP001095.1). (C) NagR-binding motif in *B. longum* subsp. *infantis* ATCC 15697 based on 11 predicted operators.

and potentially other HMO utilization in this bacterium. This conjecture was corroborated by transcriptome profiling of the *nagR* knockout mutant and by direct assessment of the binding of recombinant NagR to its predicted operators. The inferred NagR regulon structure in *B. longum* subsp. *infantis* indicates that this microorganism is adapted to simultaneous

**TABLE 1** Composition of the reconstructed NagR regulon in *B. longum* subsp. *infantis* ATCC 15697

| Locus tag | Gene | Annotation | Classification[a] | Reference |
|-----------|------|------------|----------------|-----------|
| Blon_0879 | *nagK* | Putative ROK family *N*-acetylglucosamine kinase | EC 2.7.1.59 | |
| Blon_0880 | *nagR* | Transcriptional regulator of LNB/GNB and HMO utilization | ROK family | This study |
| Blon_0881 | *nagB* | Glucosamine-6-phosphate deaminase | EC 3.5.99.6 | |
| Blon_0882 | *nagA* | *N*-Acetylglucosamine-6-phosphate deacetylase | EC 3.5.1.25 | |
| Blon_0883 | *gltF* | LNB/GNB ABC transporter substrate-binding protein | TC 3.A.1.1 | 39 |
| Blon_0884 | *gltG* | LNB/GNB ABC transporter permease protein 1 | TC 3.A.1.1 | |
| Blon_0885 | *gltH* | LNB/GNB ABC transporter permease protein 2 | TC 3.A.1.1 | |
| Blon_2171 | *lnpD* | UDP-hexose 4-epimerase | EC 5.1.3.2 | 59 |
| Blon_2172 | *lnpC* | UTP-hexose-1-phosphate uridylyltransferase | EC 2.7.7.10 | 59 |
| Blon_2173 | *lnpB* | *N*-Acetylhexosamine 1-kinase | EC 2.7.1.162 | 59 |
| Blon_2174 | *lnpA* | 1,3-*β*-Galactosyl-*N*-acetylhexosamine phosphorylase | EC 2.4.1.211 | 63 |
| Blon_2175 | *gltC* | LNT/LNB/GNB ABC transporter permease protein 2 | TC 3.A.1.1.48 | |
| Blon_2176 | *gltB* | LNT/LNB/GNB ABC transporter permease protein 1 | TC 3.A.1.1.48 | |
| Blon_2177 | *gltA* | LNT/LNB/GNB ABC transporter substrate-binding protein | TC 3.A.1.1.48 | 39 |
| Blon_2331 | *lacS2* | Lactose MFS permease-2 | TC 2.A.2 | |
| Blon_2332 | *lacS* | Lactose MFS permease-1 | TC 2.A.2 | 84 |
| Blon_2341 | - | Hypothetical protein | | |
| Blon_2342 | *hmoC2* | Type II HMO ABC transporter permease protein 2 | TC 3.A.1.1 | |
| Blon_2343 | *hmoB2* | Type II HMO ABC transporter permease protein 1 | TC 3.A.1.1 | |
| Blon_2344 | *hmoA2* | Type II HMO ABC transporter-2 substrate-binding protein | TC 3.A.1.1 | 39 |
| Blon_2345 | *hmoC* | Type II HMO ABC transporter permease protein 2 | TC 3.A.1.1 | |
| Blon_2346 | *hmoB* | Type II HMO ABC transporter permease protein 1 | TC 3.A.1.1 | |
| Blon_2347 | *hmoA* | Type II HMO ABC transporter-1 substrate-binding protein | TC 3.A.1.1 | 39 |
| Blon_2348 | *nanH2* | HMO cluster *α*-2,3/6-sialidase | GH33 | 41 |
| Blon_2349 | *nanA2* | *N*-Acetylneuraminate lyase-2 | EC 4.1.3.3 | |
| Blon_2350 | *hmoA3* | Putative HMO ABC transporter-1 substrate-binding protein | TC 3.A.1.1 | 39 |
| Blon_2351 | *hmoA4* | Putative HMO ABC transporter-2 substrate-binding protein | TC 3.A.1.1 | 39 |
| Blon_2352 | *hmoA5* | Putative HMO ABC transporter-3 substrate-binding protein | TC 3.A.1.1 | 39 |
| Blon_2354 | *hmoA6* | Putative HMO ABC transporter-4 substrate-binding protein | TC 3.A.1.1 | 39 |

[a]Transporter Classification Database (TC) numbers for transporters, the CAZy family for GHs, Enzyme Commission (EC) numbers for downstream catabolic enzymes, and the TF family for transcription factors are shown.

utilization of multiple HMOs, suggesting the use of rationally formulated HMO mixtures rather than individual oligosaccharides as prebiotics. The reconstructed NagR regulons also provide insights into the evolution of complex regulatory networks controlling carbohydrate metabolism in bifidobacteria.

## RESULTS

**Genomic reconstruction reveals the complexity of the NagR regulon in *B. longum* subsp. *infantis*.** We used a position weight matrix (PWM)-based approach to reconstruct the NagR regulon in *B. longum* subsp. *infantis* ATCC 15697 and identified 11 potential NagR-binding sites (operators) in promoter regions of genes/operons encoding components of HMO utilization pathways, including six previously unknown NagR operators within in the H1 cluster (Fig. 1A and B and see Table S3 in the supplemental material). Among the new putative NagR regulon members were genes encoding (i) LNnT (type II HMO) ABC transporters (HmoABC, HmoA2B2C2), (ii) substrate-binding components of ABC transporters possibly involved in HMO uptake (HmoA3, HmoA4, HmoA5, HmoA6), (iii) *N*-acetylneuraminate lyase NanA2 and GH33 family *α*-sialidase NanH2, and (iv) lactose permeases (LacS, LacS2) (Fig. 1A and Table 1). Overall, the reconstructed regulon contained 29 genes. The inferred NagR-binding motif had a palindrome structure (Fig. 1C), a common feature of ROK family transcriptional regulators (51).

ROK family TFs can function as transcriptional activators or repressors (51). To infer the possible mode of action of NagR, we analyzed the position of predicted NagR operators relative to −35 and −10 promoter elements recognized by bacterial RNA polymerase holoenzyme. Ten of 11 predicted NagR operators overlapped with either −10 or −35 sequences (Fig. S1), suggesting that NagR was a potential transcriptional repressor. Therefore, based on the genomic reconstruction, we hypothesized that NagR was a global negative regulator of gene clusters involved in LNB/GNB, LNT, and LNnT utilization in *B. longum* subsp. *infantis* ATCC 15697.

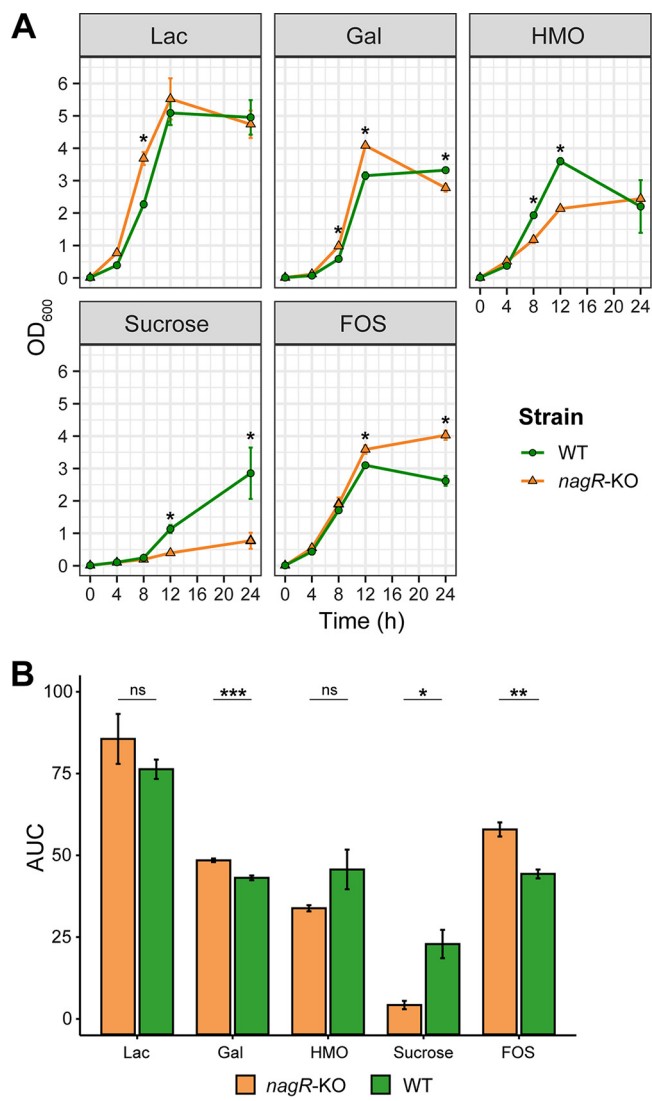

**FIG 2** Growth of *B. longum* subsp. *infantis* ATCC 15697 WT and *nagR*-KO strains in MRS-CS supplemented with various carbon sources (1% [wt/vol]). (A) Growth curves obtained by measuring $OD_{600}$ at specific time points. Data represent the mean ± SD from three biological replicates. Time points where the $OD_{600}$ values for WT and *nagR*-KO strains were significantly different (*, $P_{adj} < 0.05$) were identified using linear regression. Bonferroni correction was used to adjust for multiple testing. (B) Empirical AUC calculated by integrating the areas under growth curves. Data represent the mean ± SD from three biological replicates. Means were compared via Student's *t* test: ns, not significant ($P > 0.05$); *, $P ≤ 0.05$; **, $P ≤ 0.01$; ***, $P ≤ 0.01$.

**Engineered *nagR* insertional mutant displays comparable yet distinct physiological properties compared to the parental wild-type strain.** To experimentally study the proposed regulatory role of NagR, we generated a *nagR* knockout (*nagR*-KO) mutant of *B. longum* subsp. *infantis* ATCC 15697 by insertional mutagenesis and verified the insertion position using genomic PCR (Fig. S2A and B). The Sp$^r$ gene insertion at the *nagR* locus was maintained in the *nagR*-KO genome for at least 30 generations without antibiotic pressure (Fig. S2C). We used empirical area under the curve (AUC) to integrate information from growth curves of *nagR*-KO and wild-type (WT) strains cultivated in MRS-CS medium (see Materials and Methods) supplemented with various carbon sources (Fig. 2A). The AUC values for *nagR*-KO and WT strains grown in MRS-CS-Lac or MRS-CS-HMO were not significantly different (Fig. 2B). In contrast, the *nagR*-KO mutant had significantly lower AUC when grown in MRS-CS supplemented with sucrose and significantly higher AUC in the medium containing fructooligosaccharides (FOS) (Fig. 2B).

HMO consumption profiling of the WT and *nagR*-KO mutant grown in MRS-CS-HMO revealed that both strains completely salvaged Lac, LNT, LNnT, 2'/3-fucosyllactose (FL),

and difucosyllactose (DFL) after 24 h (Fig. S3A). Compared to the WT strain, the *nagR*-KO mutant consumed more LNT at the 4- and 8-h time points and displayed delayed consumption of large fucosylated HMOs, namely, lacto-*N*-fucopentaoses (LNFP) I/II/III, and lacto-*N*-difucohexaoses (LNDFH) I/II (Fig. S3A). The mutant also expelled less Fuc into the medium (Fig. S3A). Metabolic profiling of supernatants of the WT and *nagR*-KO strains cultivated in MRS-CS-HMO revealed that both strains released similar quantities of acetate (Fig. S3B). However, the *nagR*-KO mutant produced significantly more formate and less lactate at the 8-, 12-, and 24-h time points (Fig. S3B). These data demonstrated comparable yet distinct physiological properties of WT and *nagR*-KO strains, prompting a follow-up transcriptome profiling to infer genes differentially expressed in the mutant.

**Comparative transcriptomics corroborates the role of NagR as a global negative regulator of LNB/GNB, LNT, and LNnT utilization in *B. longum* subsp. *infantis*.** We used transcriptome sequencing (RNA-seq) to compare transcriptomes of *nagR*-KO and WT strains of *B. longum* subsp. *infantis* ATCC 15697 grown in MRS-CS supplemented with Lac or LNnT, yielding four experimental conditions (Fig. 3A). The choice of carbon sources was based on a previous study, where LNnT had induced many HMO utilization genes and Lac had been used as a comparator (45). Principal component analysis (PCA) of TMM-normalized (52) counts revealed that each experimental condition formed a distinct cluster (Fig. 3B). Specifically, PCA separated samples by carbon source (principal component 1 [PC1], 38.8% of the total variance) and strain (PC2, 21.1% of the total variance).

Linear modeling implemented in the *limma* framework (53) revealed significant upregulation (fold change [FC] of >2 and adjusted *P* [$P_{adj}$] value of <0.01) of multiple *nag*, *lnp*, and H1 cluster genes in the *nagR*-KO strain grown in MRS-CS-Lac compared to the WT grown in MRS-CS-Lac (Fig. 3C and Table S2A). Overall, 19 out of 29 genes constituting the reconstructed NagR regulon were upregulated. These results demonstrate that NagR, in line with the bioinformatic prediction, functions as a global transcriptional repressor of the *nag*, *lnp*, and H1 loci in *B. longum* subsp. *infantis* ATCC 15697. In addition, we observed significant upregulation of the *malEFG* operon encoding an ABC transport system for maltose (39) and significant downregulation (FC <−2 and $P_{adj}$ < 0.01) of *cscA*, *cscB*, and *cscR* in the *nagR*-KO mutant (Fig. 3C and Table S2A). The latter genes are involved in sucrose uptake and catabolism (54) and are predicted to be controlled by a local repressor from the LacI family, CscR (48). The downregulation of the *csc* genes was consistent with significantly decreased AUC of the *nagR*-KO mutant cultivated in the medium supplemented with sucrose (Fig. 2B). No potential NagR operators were identified in the promoter regions of *malEFG* and *csc* cluster genes, suggesting that the effect of the *nagR* knockout on their expression was indirect.

RNA-seq of the WT strain revealed that 25 out of 29 genes constituting the predicted NagR regulon were upregulated during growth in MRS-CS-LNnT compared to MRS-CS-Lac (Fig. 3D and Table S2B). The expression of *nag*, *lnp*, and H1 cluster genes was higher in the WT strain cultured in the presence of LNnT than in the *nagR*-KO mutant cultured in MRS-CS-Lac (Fig. S4). These observations point to the presence of additional transcriptional activation mechanism(s) that may contribute to the upregulation of these gene clusters during the growth of *B. longum* subsp. *infantis* in the medium containing LNnT. In addition, we observed a significant upregulation of *malK* (Fig. S4 and Table S2B), encoding a shared ATPase component that potentially energizes multiple carbohydrate-specific ABC transporters in *B. longum* subsp. *infantis* (48). The upregulation of *malK* is likely tied with the enhanced energetic needs associated with the induction of genes encoding multiple ABC transport systems (GltFGH, GltABC, HmoABC, HmoA2B2C2, HmoA3 to A5).

Overall, the obtained transcriptomic data are consistent with the bioinformatic regulon reconstruction and demonstrate that NagR regulates LNB/GNB, LNT, and LNnT utilization in *B. longum* subsp. *infantis* by repressing genomic loci encoding (i) transporters of respective glycans and (ii) GlcNAc and LNB/GNB catabolic pathways.

**Interaction of NagR with predicted operators is dependent on GlcNAc and its phosphorylated derivatives.** To test the interaction between NagR and its predicted binding sites, we cloned the *nagR* gene from *B. longum* subsp. *infantis* ATCC 15697 and expressed the recombinant protein as a fusion with an N-terminal His tag in *Escherichia coli* BL21(DE3). The electrophoretic mobility shift assay (EMSA) demonstrated that NagR

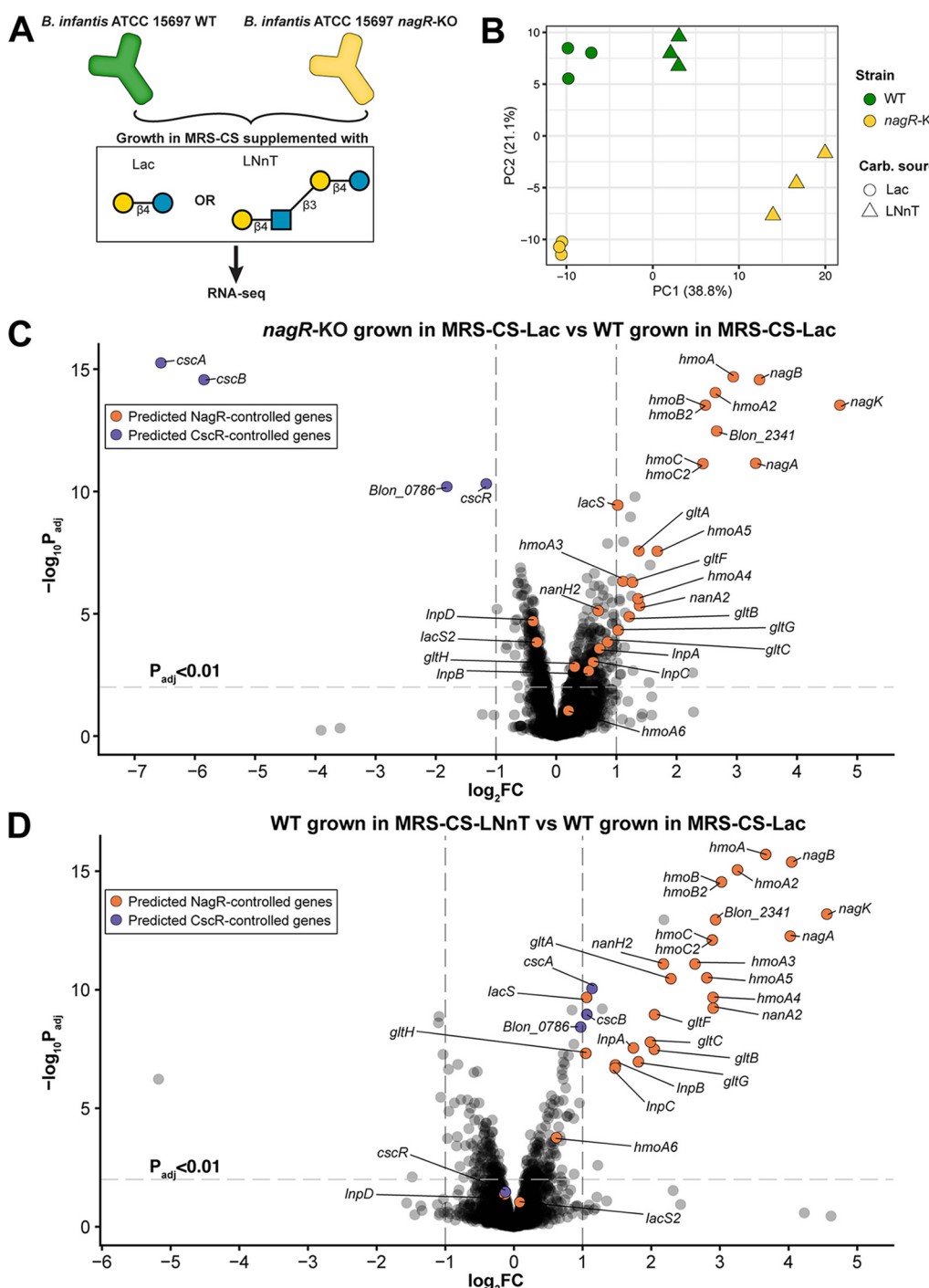

**FIG 3** RNA-seq of WT and *nagR*-KO strains grown in MRS-CS supplemented with Lac or LNnT. (A) Schematic representation of the experimental design; (B) PCA of TMM-normalized count data. Each data point represents one sample. (C and D) Volcano plots depicting the $\log_2$ FC of gene expression versus the $-\log_{10} P_{adj}$. Panel C compares *nagR*-KO and WT strains grown in MRS-CS-Lac, whereas panel D compares the WT strain grown in MRS-CS-LNnT and MRS-CS-Lac. Criteria for calling differentially expressed genes were as follows: $P_{adj} < 0.01$ and absolute FC > 2. Genes constituting the reconstructed NagR and CscR regulons are colored in orange and purple, respectively.

specifically bound DNA fragments (probes) containing candidate NagR operators located upstream of *nagK*, *nagB*, *gltA*, *hmoA*, *hmoA2*, and *hmoA3* genes (Fig. 4A and B and Fig. 5). Titration with increasing concentrations of NagR revealed that probes *nagK*, *nagB_I*, *hmoA*, *hmoA2* had a high affinity for NagR (50% effective concentration [EC$_{50}$], 13 to 23 nM), whereas probes *gltA* and *hmoA3* had a moderate one (EC$_{50}$, 140 to 180 nM) (Fig. 5). Reactions

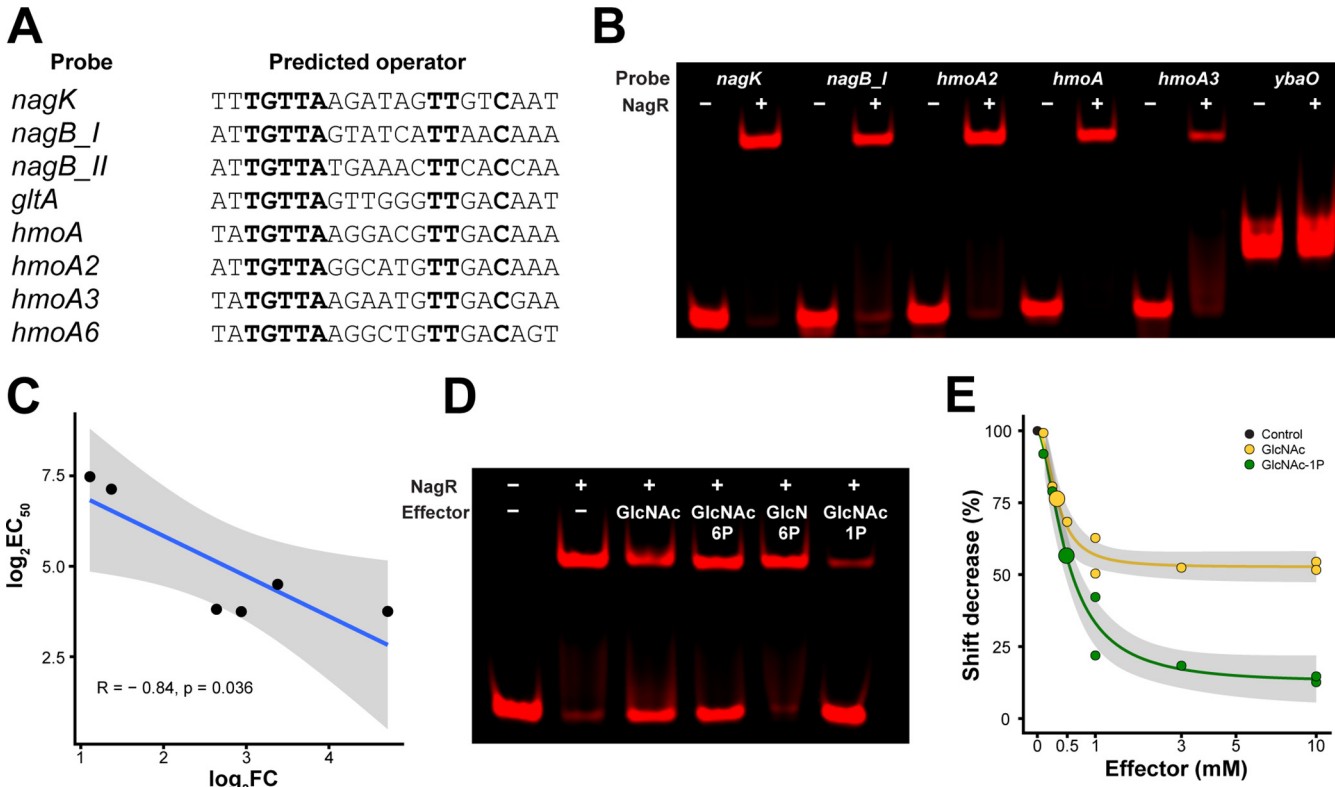

**FIG 4** Interactions of recombinant NagR with predicted operators and screening of NagR effectors. (A) Predicted NagR operators identified in the promoter regions of listed genes. Conserved nucleotides are in boldface. The full sequences of probes used in EMSAs are given in Table S1. (B) EMSA gel showing interactions of NagR with DNA probes containing predicted operators. The NagR concentration was 25 nM, and probe concentrations were 1 nM. The *ybdO* probe was used as a control for nonspecific binding. (C) Correlation between probe EC$_{50}$ values determined via EMSA and expression FCs of cognate genes in the RNA-seq experiment (*nagR*-KO mutant versus WT grown in MRS-CS-Lac). (D) Effect of various GlcNAc metabolism intermediates (10 mM) on the interaction between NagR (25 nM) and the *hmoA* probe (1 nM). (E) EC$_{50}$ values of selected NagR effector molecules. The *y* axis depicts the ratio % shift with effector/% shift without effector. Calculated EC$_{50}$ values are shown as big circles. Concentrations of NagR and the *hmoA* probe were 25 nM and 1 nM, respectively.

with probes *nagB_II* and *hmoA6* did not manifest in robust shifts even at high (>500 nM) protein concentrations (Fig. S5A). These results demonstrate that NagR exhibits different affinities to its various operators. We also observed a significant negative correlation (Pearson $R = -0.84$, $P < 0.05$) between calculated EC$_{50}$ values for probes and expression FCs for corresponding genes in the RNA-seq experiment (*nagR*-KO mutant versus WT grown in MRS-CS-Lac) (Fig. 3C and 4C), indicating the more tightly NagR bound to an operator, the stronger was the observed upregulation of a respective gene in the *nagR*-KO strain.

To identify potential NagR effector molecules, we used a probe with a high affinity to NagR (*hmoA*) and added various GlcNAc metabolism intermediates (GlcNAc, GlcNAc-6P, GlcN-6P, GlcNAc-1P) to the binding reaction mixture. GlcNAc, GlcNAc-6P, and GlcNAc-1P, but not GlcN-6P, disrupted the NagR-DNA complex at a saturating (10 mM) concentration (Fig. 4D). Titration of the effectors revealed that GlcNAc-6P lost its complex-disrupting effect at 1 mM, whereas GlcNAc and GlcNAc-1P lost their effect at 0.1 mM (Fig. S5B). The calculated effector EC$_{50}$ values were 0.33 ± 0.05 mM for GlcNAc and 0.49 ± 0.06 mM for GlcNAc-1P (Fig. 4E).

Overall, the EMSA results demonstrate that (i) NagR binds its predicted operators with various affinities, and (ii) multiple GlcNAc metabolism intermediates disrupt NagR-DNA interactions *in vitro* and thus serve as plausible NagR transcriptional effectors in *B. longum* subsp. *infantis*.

**The NagR regulon was a subject of evolutionary expansion in *Bifidobacteriaceae*.** We used the PWM-based approach to identify NagR binding motifs and reconstruct regulons in 25 genomes representing various phylogenetic lineages within the *Bifidobacteriaceae* family to trace the potential evolutionary history of this gene regulatory network. Overall, the sizes and compositions of the reconstructed NagR regulons markedly varied among the studied genomes (Fig. 6A and Table S3). For instance, early diverged *Bifidobacterium*

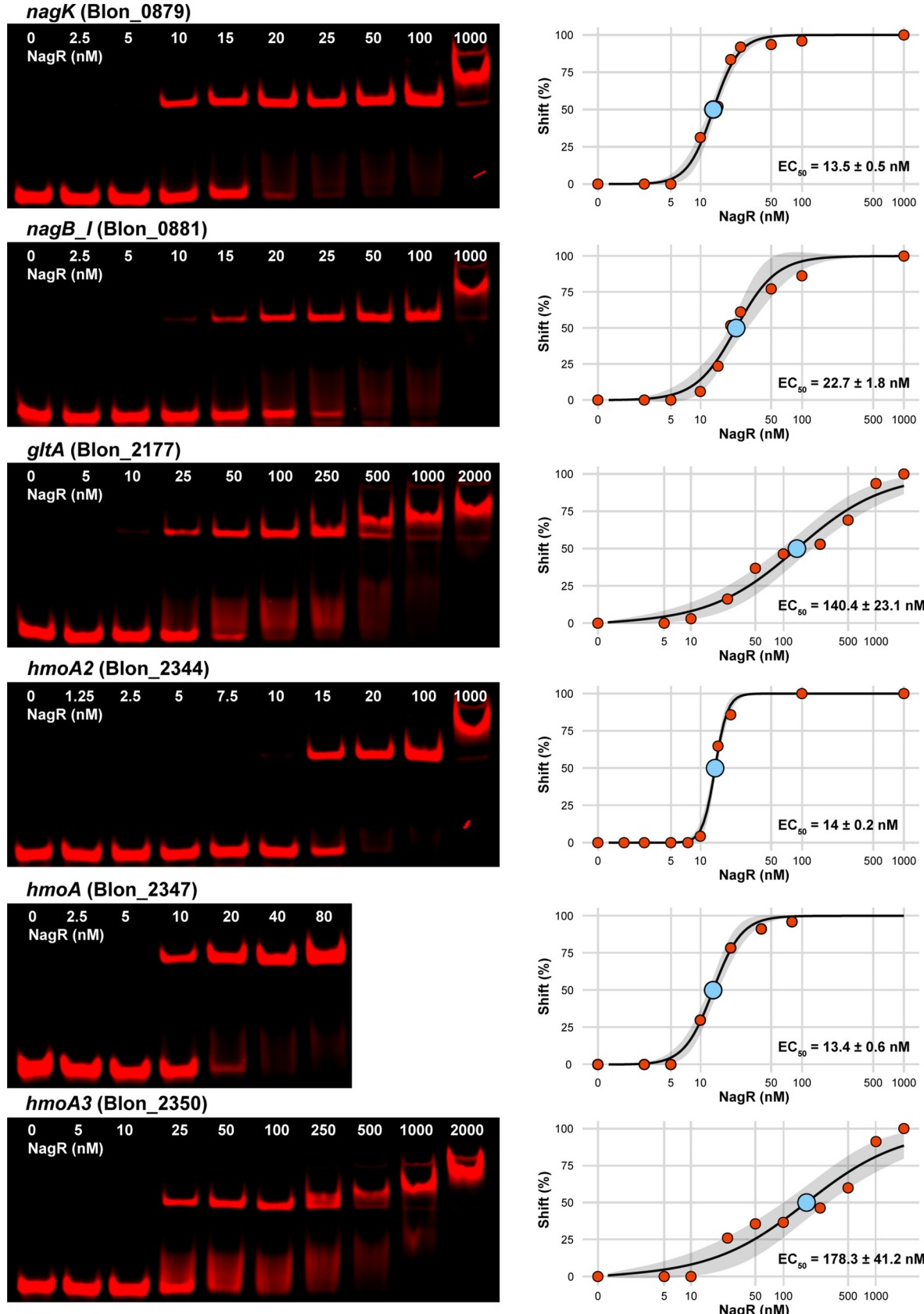

**FIG 5** EMSA gels depicting titration of DNA probes (1 nM) containing predicted NagR operators with recombinant NagR. Gels were quantified, and the results were approximated by the 4PL equation. The NagR concentration at which half of the probe is shifted ($EC_{50}$) is shown. Gray shading depicts 95% confidence intervals. The $x$ axis is in the $log_{10}$ scale.

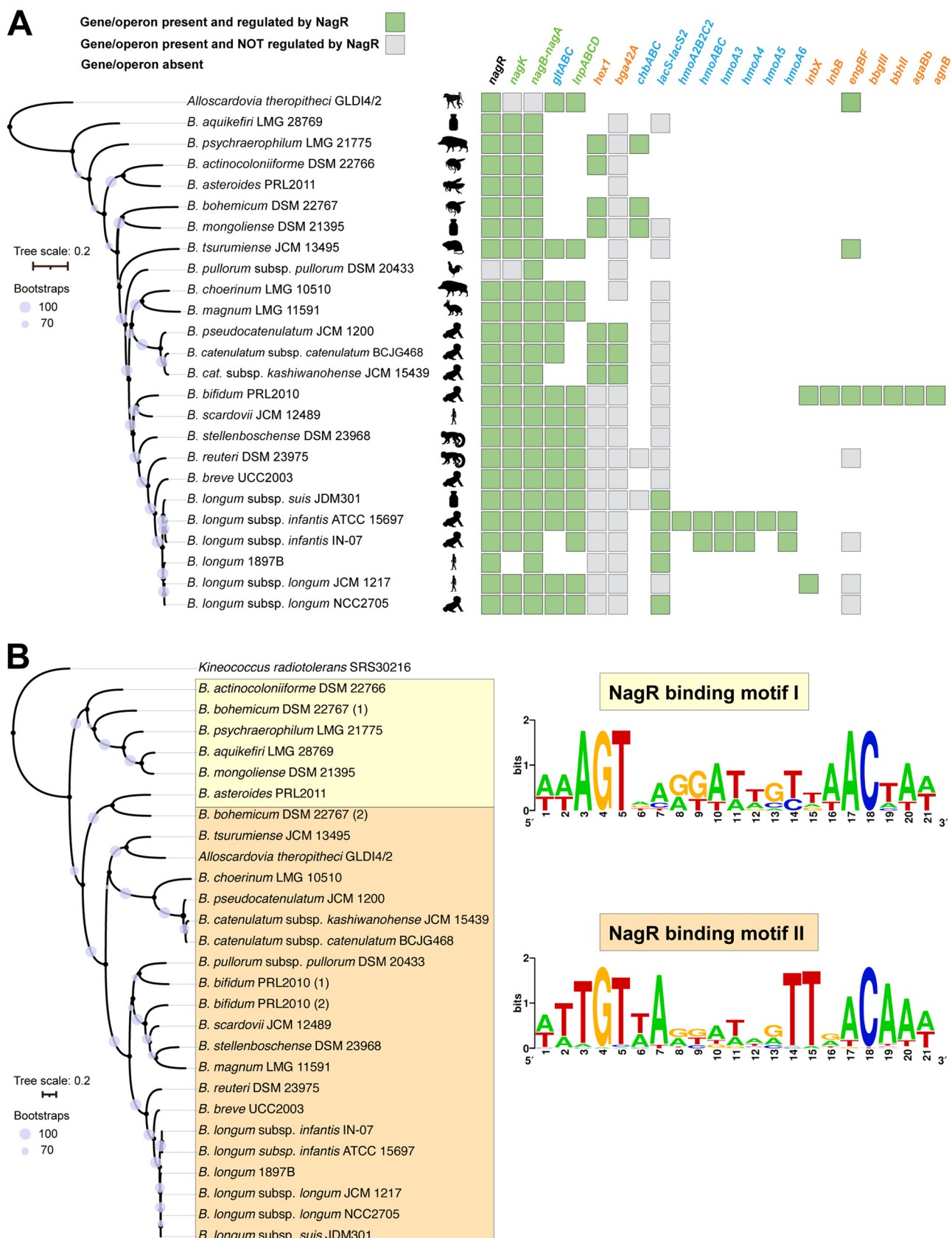

**FIG 6** Evolution of the NagR regulon and binding motif within the *Bifidobacteriaceae* family. (A) NagR regulon composition in 25 *Bifidobacteriaceae* strains mapped on a species tree built based on the alignment of 247 core genes. Bootstrap values are shown as purple circles. Black symbols

species isolated from insects and dairy products possessed concise NagR regulons comprised of a single genomic locus (*nag*) encoding GlcNAc and predicted *N,N'*-diacetylchitobiose catabolic pathways (Text S1). This observation suggests that in ancestral bifidobacteria, the role of NagR was likely confined to local regulation of the GlcNAc catabolic pathway.

We observed gradual NagR regulon expansion in multiple *Bifidobacterium* genomes isolated from mammalian hosts (Fig. 6A, Table S3, and Text S1). The largest and most complex NagR regulons were identified in strains isolated from the human neonatal gut, namely, *B. longum* subsp. *infantis* ATCC 15697 and *B. bifidum* PRL2010. While in *B. longum* subsp. *infantis*, the NagR regulon expanded to include multiple HMO transporters encoded within the H1 cluster (Fig. 1B), in *B. bifidum*, the reconstructed regulon contained genes encoding multiple GHs involved in the extracellular degradation of HMOs and mucin *O*-glycans (Text S1). Additionally, the NagR regulon expansion was accompanied by minor variations in the NagR-binding motif (Fig. 6B and Text S1). Taken together, the comparative regulon reconstruction suggests that NagR evolved from a local regulator of GlcNAc catabolism in ancestral bifidobacteria to a global regulator of utilization of various host glycans (e.g., HMOs) in species isolated from the mammalian neonatal gut.

## DISCUSSION

**Regulation of HMO utilization in *B. longum* subsp. *infantis*.** The predominance of bifidobacteria in the infant gut is linked to their ability to consume and use HMOs as a carbon source (25, 26, 38, 40, 55, 56). *B. longum* subsp. *infantis* possesses a unique gene cluster (H1) that encodes multiple catabolic enzymes and components of ABC transporters that endow this species with the ability to utilize a multitude of HMOs (25, 36, 38). Previous studies demonstrated that a pooled HMO mixture and individual HMOs (LNT and LNnT) induce the expression of H1 and *nag* cluster genes in *B. longum* subsp. *infantis* ATCC 15697, suggesting that H1 acts as an HMO-inducible unit and is coregulated with the GlcNAc catabolic pathway (45, 46). However, the regulatory mechanisms underlying this phenomenon were not elucidated.

In this study, we have established NagR-mediated repression of H1, *lnp*, and *nag* cluster genes in *B. longum* subsp. *infantis* ATCC 15697 by combining PWM-based regulon reconstruction with transcriptome profiling of the *nagR*-KO mutant. The composition of the NagR regulon suggests that this global TF regulates the utilization of LNB/GNB, LNT, LNnT, and potentially other decorated (e.g., sialylated) type I and II HMOs in *B. longum* subsp. *infantis*. We have also demonstrated the concentration-dependent binding of recombinant NagR to its predicted operators *in vitro*. The $EC_{50}$ values inferred from EMSAs negatively correlated with fold change values for upregulated genes in the *nagR*-KO mutant. Thus, the degree of gene repression by NagR is strongly dependent on the affinity of this TF to its cognate operators in the promoter regions of corresponding genes.

We identified GlcNAc and its phosphorylated derivatives, GlcNAc-6P and GlcNAc-1P, as potential NagR transcriptional effectors in *B. longum* subsp. *infantis* ATCC 15697. This result is somewhat unexpected since James et al. previously reported GlcNAc-6P, but not GlcNAc, as the NagR transcriptional effector in *B. breve* UCC2003 (50). This discrepancy may reflect a metabolic adaptation to the more global nature of NagR regulon in *B. longum* subsp. *infantis*, although, alternatively, it may be due to the differences in the experimental approach, which in our case was based on the use of purified recombinant NagR for EMSAs versus crude cell lysate as in the previous study (50). EMSA data indicated that the acetyl group of these intermediary metabolites played a crucial role in NagR-effector interactions, whereas phosphorylation of GlcNAc appeared to be dispensable. While GlcNAc and GlcNAc-6P have been described as transcriptional effectors of the ROK family TFs (50, 57, 58), the potential effector role of GlcNAc-1P is novel and unexpected. Although *N*-acetylhexosamine 1-kinase

**FIG 6** Legend (Continued)

indicate strain isolation sources. Regulon members are colored according to their function: catabolic enzymes are green, GHs are orange, and transporters are blue. (B) NagR-binding motifs mapped on a tree of NagR proteins from 25 *Bifidobacteriaceae* strains. Bootstrap values are shown as purple circles. NagR paralogs in *B. bohemicum* and *B. bifidum* are denoted by the numbers 1 and 2.

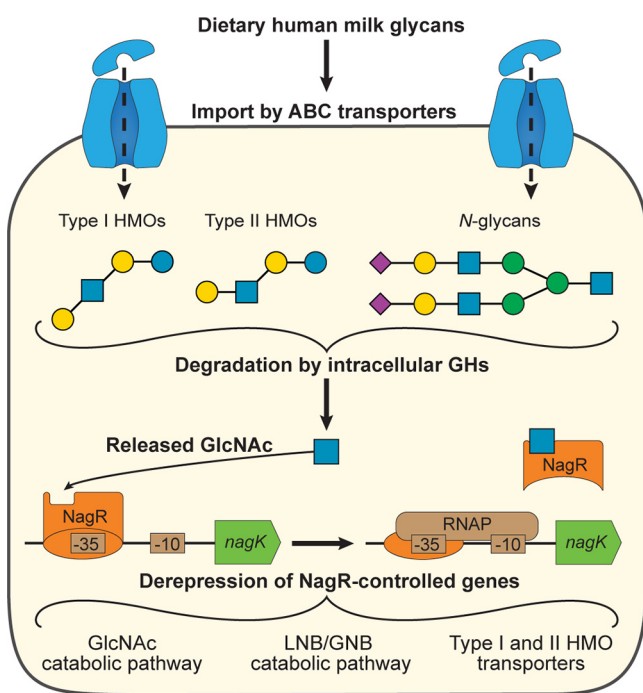

**FIG 7** Model of NagR-mediated regulation of HMO utilization in *B. longum* subsp. *infantis*. (Step 1) GlcNAc-containing milk glycans (e.g., LNT and LNnT) are taken up into the cell by various ABC transporters. (Step 2) Once inside the cell, the glycans are degraded by intracellular GHs, and GlcNAc is released. (Step 3) Released GlcNAc interacts with NagR and disrupts the NagR-operator complex, leading to derepression of NagR-controlled genes.

(LnpB) can phosphorylate GlcNAc to GlcNAc-1P (59), no enzymes that convert the latter to GlcNAc-6P and thus shunt it to the GlcNAc catabolic pathway have been described in prokaryotes (60). In contrast, GlcNAc-1P can be converted to UDP-GlcNAc by GlcNAc-1P uridyltransferase (GlmU) and enter the peptidoglycan biosynthesis pathway (61). Therefore, additional studies are required to assess the biological significance of GlcNAc-1P functioning as a potential NagR transcriptional effector.

Based on the obtained data, we propose a model (Fig. 7) in which the release of GlcNAc during degradation of LNT and LNnT by intracellular GHs results in derepression of *nag*, *lnp*, and H1 clusters, including genes encoding LNT and LNnT transporters. This model explains the similarity of transcriptomic responses of *B. longum* subsp. *infantis* to LNT or LNnT in batch cultures (45, 46) and gnotobiotic mice (15) and suggests that utilization of any GlcNAc-containing glycan by this bacterium will result in the upregulation of NagR-controlled genes. Among these glycans might be particular fucosylated HMOs (e.g., LNFP I) and milk *N*-glycans imported by other ABC transport systems (15, 40). Consistent with this notion, a previous study demonstrated that *B. longum* subsp. *infantis* upregulates *nag* and H1 cluster genes (62) when utilizing *N*-glycosylated human lactoferrin. The proposed model, however, does not explain all transcriptional responses observed during HMO utilization by *B. longum* subsp. *infantis*. The transcriptomic profile of the WT strain grown in MRS-CS-LNnT suggests that additional mechanisms may coactivate the expression of NagR regulon genes (particularly within the *lnp* cluster) under physiologically inducing conditions.

The structure of the NagR-mediated transcriptional network in *B. longum* subsp. *infantis* likely reflects the evolutionary adaptation of this bacterium to simultaneous foraging of multiple distinct HMOs and other milk glycans. This notion suggests that using a mixture of LNT and LNnT (and potentially other HMOs) rather than individual oligosaccharides as a prebiotic may be a more efficient solution for selective stimulation of *B. longum* subsp. *infantis* growth in the neonatal gut since it considers the nuanced regulatory mechanisms and physiology of the target organism.

**Evolution of the NagR regulon in bifidobacteria.** Evolution of *B. longum* subsp. *infantis* was shaped by its ecological niche, specifically adaptation to the foraging of dietary

milk glycans (e.g., HMOs) abundantly present in the gut of breastfed infants (38). This adaptation led to the emergence of several unique gene clusters, such as H1 (38), controlled by a complex NagR regulatory network. However, other *Bifidobacterium* species inhabiting the neonatal gut, such as *B. breve* and *B. longum* subsp. longum, do not harbor the H1 cluster and have less complex NagR regulons (48, 50). Thus, to elucidate the plausible evolutionary history of the NagR regulon, we reconstructed its content in 25 representative genomes spanning 18 *Bifidobacterium* species and one *Alloscardovia* species isolated from various hosts and environments.

The regulon structure in early diverged *Bifidobacterium* species (e.g., *B. asteroides* and *B. aquikefiri*) suggests that NagR potentially functioned as a local regulator of a single gene cluster involved in GlcNAc and possibly *N,N'*-diacetylchitobiose catabolism in ancestral bifidobacteria. Bifidobacteria colonizing mammalian hosts have acquired various gene clusters encoding the catabolic machinery involved in host glycan utilization. For example, most bifidobacteria isolated from mammals (including humans) harbor the *lnp* cluster encoding a transporter and a catabolic pathway for LNB and GNB, structural components of various milk (HMOs and glycolipids) and intestinal (mucin *O*-glycans) glycans, respectively (59, 63–65). The involvement of the *lnp* cluster in the reconstructed NagR regulons suggests transitioning from a local to a multilocus-controlling TF. *Bifidobacterium pseudocatenulatum* and two closely related species lacking the *lnp* genes have NagR regulons potentially expanded to control the LNB/LNT utilization machinery (*gltABC-nagK-hex1-nagB-nagA* and *bga42A* genes). The structure of the reconstructed NagR regulon in *B. pseudocatenulatum* is consistent with a recently published transcriptomic data set in which the expression of these genes was induced by LNFP I (66). Although the import of LNFP I is mediated by an ABC transport system for fucosylated HMOs (55, 66) and not GltABC, the GlcNAc released during the degradation of this oligosaccharide was likely responsible for the derepression of NagR-controlled genes.

Independent NagR regulon expansion events potentially occurred in prevalent infant-associated species with the highest HMO utilization potential: *B. longum* subsp. *infantis* and *B. bifidum*. In two closely related strains of *B. longum* subsp. *infantis*, the reconstructed regulon includes multiple HMO transporter genes from the H1 cluster, whereas in *B. bifidum*, NagR potentially regulates extracellular GHs involved in HMO and mucin *O*-glycan degradation. These observations provide a fascinating example of how catabolic machinery corresponding to two distinct strategies of HMO utilization (intracellular in *B. longum* subsp. *infantis* versus extracellular in *B. bifidum*) may have converged toward transcriptional control by the same TF. The structure of the reconstructed NagR regulon in *B. bifidum* PRL2010 partially explains the upregulation of *nag*, *lnp*, and genes encoding extracellular GHs during the growth of this strain in a mucin-supplemented medium (56).

Potential expansion of regulons for carbohydrate metabolism genes has been previously described in various bacterial lineages (67, 68), including bifidobacteria (69, 70). However, the underlying rationale of these expansion events to include specific genes/operons was not always straightforward. Here, we hypothesize that the NagR regulon expansion to control genes involved in the catabolism of GlcNAc-containing host glycans in *Bifidobacterium* might be linked to the ability of NagR orthologs to sense GlcNAc and/or its phosphorylated derivatives (50). Interestingly, other ROK family TFs have been implicated in regulating the utilization of GlcNAc-containing glycans in bifidobacteria. For example, in *B. breve* UCC2003, while NagR functions as a negative regulator of LNB/GNB and LNT utilization pathways, its paralog, NahR, represses a gene encoding an LNnT transporter (50). Another NagR paralog in this strain, AtsR2, represses a gene cluster involved in utilizing GlcNAc-6S, a mucin *O*-glycan-constituting saccharide released by 6-sulfo-$\beta$-*N*-acetylglucosaminidase BbhII (71, 72). Different GlcNAc derivatives were reported as effector molecules of these TFs: GlcNAc for NahR and GlcNAc-6S for AtsR2 (50, 71). Finally, a recently identified NagR paralog in *B. longum* subsp. *infantis* Bg_2D9, NglR, potentially controls a gene cluster involved in the metabolism of complex *N*-glycans (15). These data suggest that the evolution of gene regulatory networks governing the utilization of GlcNAc-containing glycans in bifidobacteria was not limited to the NagR regulon expansion. Another scenario might have involved duplication(s) of the *nagR* gene after speciation followed by the functional divergence of emerged paralogs.

Overall, these observations illustrate how bifidobacteria adapted to regulate the foraging of host glycans during the colonization of mammalian hosts and shed more light on how complex regulatory networks emerge and evolve.

## MATERIALS AND METHODS

**Reagents and bacterial strains.** Reagents were purchased from Alfa-Aesar (Tewksbury, MA, USA), Ambion (Austin, TX, USA), Combi-Blocks (San Diego, CA, USA), Sigma-Aldrich (St. Louis, MO, USA), and Invitrogen (Carlsbad, CA, USA), unless indicated otherwise. Synthetic LNnT (>95% purity) was generously donated by DSM (Heerlen, Netherlands). The HMO mixture was prepared from pooled human milk (55). The experimental protocol was reviewed and approved by the Ethics Committee of Kyoto University (R0046); the study was performed per the Declaration of Helsinki, and informed consent was obtained from all mothers (all subjects). Oligonucleotides were synthesized by Integrated Genomic Technologies (Coralville, IA, USA). Phusion high-fidelity DNA polymerase, restriction enzymes, and Quick Ligase were purchased from New England BioLabs (Ipswich, MA, USA).

The type strain of *B. longum* subsp. *infantis* (ATCC 15697 = JCM 1222) was obtained from Japan Collection of Microorganisms (RIKEN BioResource Research Center, Tsukuba, Japan). *Escherichia coli* DH5$\alpha$ and One Shot TOP10 (Invitrogen) cells were used for genetic manipulations. *E. coli* BL21(DE3) (New England BioLabs) was used for recombinant NagR overexpression.

**Bioinformatic analysis.** A previously established comparative genomics approach was used to identify putative NagR binding sites and reconstruct regulons in *B. longum* subsp. *infantis* ATCC 15697 and other selected strains (48, 69). For the initial NagR regulon reconstruction, we built a PWM based on data available in the RegPrecise database (48, 73) using SignalX (74). To improve the identification of NagR operators, we built additional PWMs representing two different NagR-binding motifs specific for distant bifidobacterial lineages. Constructed PWMs were used to search for new potential NagR operators using GenomeExplorer (74) with the following parameters: (i) positions −500 to +50 bp relative to the first codon of a gene and (iii) site score threshold of 4.3. Identified sites were screened using the consistency check and phylogenetic footprinting approaches to filter out false positives (75). NagR-binding motifs were visualized via WebLogo (76). Positions of the −10 and −35 promoter elements were determined via similar PMW-based searches based on data available for *B. breve* (77) and *B. longum* (78). Promoter regions were aligned using Pro-Coffee (79). Details on additional genomic analysis of *Bifidobacterium* strains and phylogenetic inference are available in Text S1 in the supplemental material.

**Targeted *nagR* gene disruption in *B. longum* subsp. *infantis* ATCC 15697.** A single-crossover recombination event was used to inactivate the *nagR* gene (Blon_0880; UniProt entry B7GQA0) in *B. longum* subsp. *infantis* ATCC 15697. Briefly, a BamHI-digested, 2.0-kb fragment of pBS423 (80) that carries the pUC ori and a spectinomycin resistance (Sp$^r$) gene was self-ligated to generate pTK2051, a plasmid incapable of replicating in bifidobacteria. The internal region of *nagR* was then amplified by PCR using primers NagR_I/NagR_II (Table S1) and genomic DNA as a template. The amplified 0.5-kbp fragment was inserted into the BamHI site of pTK2051 using the In-Fusion Snap Assembly kit (TaKaRa Bio USA, Mountain View, CA, USA). The resulting suicide plasmid was introduced into *B. longum* subsp. *infantis* by electroporation (40). To prepare electrocompetent cells, *B. longum* subsp. *infantis* ATCC 15697 was grown in 50 mL of Gifu anaerobic medium (GAM; Nissui Pharmaceutical, Tokyo, Japan) to an optical density at 600 nm (OD$_{600}$) of 0.2. Cells were harvested by centrifugation (4,800 × *g* for 15 min at 4℃), washed with ice-cold 1 mM ammonium citrate buffer (pH 6.0) containing 50 mM sucrose, and resuspended in 400 $\mu$L of the same buffer. An aliquot (200 $\mu$L) was mixed with 10 $\mu$g of the suicide plasmid and then pulsed using a Gene Pulser Xcell system (Bio-Rad Laboratories, Hercules, CA, USA) with 10 kV/cm, 25 $\mu$F, and 200 Ω. The pulsed cells were immediately mixed with 800 $\mu$L of 1% (wt/vol) Lac-supplemented GAM and incubated at 37℃ under anoxic conditions for 3 h before being spread on GAM agar plates containing 1% (wt/vol) Lac and 15 $\mu$g/mL spectinomycin (Sp). Colonies that appeared on the plates were subsequently subjected to a genomic PCR analysis at the *nagR* locus using primers NagR_III/NagR_IV (Table S1 and Fig. S2A). The amplicon was directly sequenced to ensure that the suicide plasmid was integrated into the intended site. The stability of the Sp$^r$ gene insertion in the absence of antibiotic pressure was additionally monitored (Text S1).

**Culture conditions.** *B. longum* subsp. *infantis* ATCC 15697 was routinely grown in GAM or Lactobacilli MRS broth without dextrose (Alpha Biosciences, Baltimore, MD, USA) with 0.34% (wt/vol) sodium ascorbate and 0.029% (wt/vol) L-cysteine–HCl monohydrate (MRS-CS). The MRS-CS medium was supplemented with Lac, LNnT, or a mixture of neutral HMOs at a final concentration of 1% (wt/vol). Cultures were incubated at 37℃ in an AnaeroPack system (Mitsubishi Gas Chemical Company, Tokyo, Japan) or a chamber maintained with a gas mix of 10% H$_2$, 10% CO$_2$, and 80% N$_2$ (Coy Laboratory Products, Grass Lake, MI, USA). Growth was monitored by measuring either culture turbidity in McFarland units using a DEN-1B densitometer (Grant Instruments, Shepreth, United Kingdom) or optical density at 600 nm (OD$_{600}$) using a DU800 spectrophotometer (Beckman Coulter, Brea, CA, USA). *E. coli* strains were cultured in Luria-Bertani broth at 37℃ with vigorous agitation. Where appropriate, growth media were supplemented with spectinomycin (15 $\mu$g/mL for *B. longum* subsp. *infantis* ATCC 15697 *nagR*-KO, 75 $\mu$g/mL for *E. coli* DH5$\alpha$) or kanamycin (50 to 60 $\mu$g/mL for all other *E. coli* strains). Details on measuring the growth, HMO consumption, and organic acid production are described in Text S1.

**Transcriptome analysis.** Overnight cultures of *B. longum* subsp. *infantis* ATCC 15697 WT and *nagR*-KO strains grown in MRS-CS-Lac (with Sp in the case of *nagR*-KO) were harvested, washed with sugar-free MRS-CS, and used to inoculate antibiotic-free MRS-CS medium supplemented with either Lac or LNnT (1% [wt/vol]) at an OD$_{600}$ of 0.02. Samples (2 mL, biological triplicates) were collected at the early to mid-exponential phase (OD$_{600}$ = 0.35) and immediately pelleted in a prechilled centrifuge at 4,800 × *g* for 5 min. Cell pellets were snap-frozen in liquid nitrogen and stored at −80℃ until further use. RNA was extracted as described previously (81)

with minor modifications; the detailed protocol can be found in Text S1. rRNA was depleted with the NEBNext rRNA depletion kit for bacteria (New England Biolabs). Barcoded libraries were made with NEBNext Ultra II directional RNA library prep kit for Illumina (New England Biolabs). Libraries were pooled and sequenced (single-end 75-bp reads) on Illumina NextSeq 500 using the High Output V2 kit (Illumina, San Diego, CA, USA). Sequencing data were analyzed as described previously (82) with certain modifications; the details are described in Text S1.

**Cloning, expression, and purification of recombinant NagR.** Codon-optimized nucleotide sequence of *nagR* (Blon_0880) was synthesized by GeneArt gene synthesis (Thermo Fisher Scientific, Waltham, MA, USA), PCR amplified using primers NagR_HisN_F and NagR_HisN_R (Table S1), digested by BamHI and SalI, and ligated into a predigested in-house pET-49b(+) vector conferring resistance to kanamycin. The ligation mixture was introduced into *E. coli* One Shot TOP10 cells by chemical transformation, and transformants were then selected based on kanamycin resistance. The recombinant NagR was expressed as a fusion with the N-terminal His tag under the control of a T7 promoter in *E. coli* BL21(DE3). Cells were grown in LB medium (50 mL) at 37°C to an $OD_{600}$ of ~0.6 and then transferred to 16°C. Protein expression was induced by adding 0.2 mM IPTG (isopropyl-$\beta$-$D$-thiogalactopyranoside). Cells were grown at 16°C overnight and collected by centrifugation at 4,800 × *g* for 15 min. Harvested cells were resuspended in a lysis buffer containing 10 mM HEPES buffer (pH 7.0), 100 mM NaCl, 0.15% Brij-35, and 5 mM $\beta$-mercaptoethanol. Cells were lysed by a freeze-thaw cycle, followed by sonication using Misonix sonicator 3000 (Misonix, Inc., Farmingdale, NY, USA). The cell debris was removed; the soluble fraction was loaded onto a Ni-nitrilotriacetic acid (NTA) agarose minicolumn (0.2 mL) (Qiagen, Hilden, Germany). The column was washed with 10 column volumes of At buffer (50 mM Tris-HCl [pH 8.0], 500 mM NaCl, 20 mM imidazole, 0.3% Brij-35, 5 mM $\beta$-mercaptoethanol) and 10 column volumes of At buffer with 1 M NaCl. Captured proteins were eluted with 0.6 mL of At buffer with 300 mM imidazole. The eluted protein fraction was concentrated and buffer exchanged into 10 mM Tris-HCl (pH 8.0) with 50 mM NaCl using 30-kDa Amicon Ultra 0.5-mL centrifugal filters (Millipore Sigma, Burlington, MA, USA). The protein concentration was determined by a Qubit protein assay kit (Invitrogen).

**Electrophoretic mobility shift assay.** Oligonucleotides containing predicted 21-bp NagR operators and surrounding genomic regions (14 bp from each end) were synthesized by Integrated DNA Technologies. The DNA fragment sequences, sizes, and labels used for testing are given in Table S1. Double-stranded labeled DNA probes were obtained by annealing IRD700-labeled oligonucleotides with unlabeled complementary oligonucleotides (ratio of 1:5) in a mixture of 4 mM Tris-HCl (pH 8.0), 20 mM NaCl, and 0.4 mM EDTA in a Mastercycler PRO thermal cycler (Eppendorf) overnight. Binding reactions were carried out with a final volume of 20 $\mu$L in binding buffer containing 10 mM Tris-HCl (pH 7.5), 50 mM KCl, 5 mM $MgCl_2$, 2.25 mM dithiothreitol (DTT), 0.125% Tween 20, and 2.5% glycerol. DNA probes (1 nM) were incubated with increasing concentrations of the purified NagR (0 to 2,000 nM) for 60 min at room temperature. Reaction mixtures were loaded on a Novex 6% DNA retardation gel (Thermo Fisher Scientific) and run in 0.5× Tris-borate-EDTA buffer (Thermo Fisher Scientific) at 100 V and room temperature for 45 min in XCell SureLock minicell electrophoresis system (Thermo Fisher Scientific). Gels were visualized using Odyssey CLx (Li-COR Biosciences, Lincoln, NE, USA). Bands were quantified in Image Studio v5.2 (Li-COR Biosciences). The resulting data were imported into R and approximated by a 4PL equation in the *drc* package (83) to calculate $EC_{50}$ values. In the 4PL model, the lower limit was fixed at 0 and the upper limit at 1. To identify possible NagR effectors, binding reactions were carried out with 25 nM NagR and the addition of 0.1 to 10 mM GlcNAc or its phosphorylated derivatives (GlcNAc-6P, GlcN-6P, GlcNAc-1P). $EC_{50}$ values for effectors were calculated using the 4PL equation, with the upper limit fixed at 1.

**Data availability.** The RNA-seq data set has been deposited in the Gene Expression Omnibus under accession no. GSE196064. Raw EMSA gel quantification, growth, HMO consumption, and metabolic profiling data are available on GitHub (https://github.com/Arzamasov/NagR_manuscript). Code detailing the data analysis steps is available on GitHub and in Text S2.

## SUPPLEMENTAL MATERIAL

Supplemental material is available online only.

**TEXT S1**, DOCX file, 0.1 MB.
**TEXT S2**, PDF file, 0.7 MB.
**FIG S1**, PDF file, 0.2 MB.
**FIG S2**, PDF file, 2.3 MB.
**FIG S3**, PDF file, 0.1 MB.
**FIG S4**, TIF file, 1.6 MB.
**FIG S5**, TIF file, 1.6 MB.
**TABLE S1**, PDF file, 0.1 MB.
**TABLE S2**, XLSX file, 0.02 MB.
**TABLE S3**, XLSX file, 0.02 MB.

## ACKNOWLEDGMENTS

We thank Semen Leyn (SBP) for help with computational analyses, Kazi Ahsan (Washington University in St. Louis) for guidance in RNA isolation experiments, Kang Liu and Brian James (SBP Genomics Core) for library preparation and sequencing, Daniel Beiting (University of Pennsylvania) for the DIY.transcriptomics course, Irina Rodionova (University of California San Diego) and Oleg Kurnasov (SBP) for guidance in protein expression and purification experiments,

Motomitsu Kitaoka (Niigata University), Junko Hirose (Kyoto Women's University), and Tadasu Urashima (Obihiro University of Agriculture and Veterinary Medicine) for technical support, RIKEN BRC, through the National BioResource Project of the MEXT/AMED, Japan, for providing pBS423, and Glycom A/S and DSM for generously providing LNnT.

This work was supported by JSPS KAKENHI (grant 21H02116 to T.K.) and the NIH (grant RO1 DK30292 to A.L.O.).

Conceptualization: A.A.A.; Data Curation, A.A.A.; Formal Analysis, A.A.A.; Investigation, A.A.A., A.N., M.S., and M.N.O.; Methodology, A.A.A., A.N., M.S., and M.N.O.; Supervision: T.K., D.A.R., and A.L.O.; Visualization: A.A.A.; Writing – Original Draft: A.A.A. with input from A.N.; Writing – Review & Editing: A.A.A., T.K., D.A.R., and A.L.O.

A.L.O. and D.A.R. are cofounders of Phenobiome, Inc., a company pursuing the development of computational tools for predictive phenotype profiling of microbial communities. Employment of M.S. and M.N.O. at Kyoto University is supported by Morinaga Milk Industry Co., Ltd. All other authors declare no conflict of interest.

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
