## [Reviewer comments · mSystems]

Human Milk Oligosaccharide Utilization in Intestinal Bifidobacteria is Governed by a Global Transcriptional Regulator NagR

Aleksandr Arzamasov, Aruto Nakajima, Mikiyasu Sakanaka, Miriam Ojima, Takane Katayama, Dmitry Rodionov, and Andrei Osterman

Corresponding Author(s): Andrei Osterman, Sanford Burnham Prebys Medical Discovery Institute

Review Timeline:

Submission Date:	April 8, 2022
Editorial Decision:	June 1, 2022
Revision Received:	July 26, 2022
Accepted:	August 23, 2022

Editor: Stephen Lindemann

Reviewer(s): Disclosure of reviewer identity is with reference to reviewer comments included in decision letter(s). The following individuals involved in review of your submission have agreed to reveal their identity: Meichen Pan (Reviewer #1)

Transaction Report:

DOI: <https://doi.org/10.1128/msystems.00343-22>

June 1, 2022

Dr. Andrei L. Osterman
Sanford Burnham Prebys Medical Discovery Institute
La Jolla

Re: mSystems00343-22 (Human Milk Oligosaccharide Utilization in Intestinal Bifidobacteria is Governed by a Global Transcriptional Regulator NagR)

Dear Dr. Andrei L. Osterman:

Thank you for submitting your manuscript to mSystems. We have completed our review and I am pleased to inform you that, in principle, we expect to accept it for publication in mSystems. However, acceptance will not be final until you have adequately addressed the reviewer comments.

Preparing Revision Guidelines

Sincerely,

Stephen Lindemann

Editor, mSystems

Journals Department
Reviewer comments:

Reviewer #1 (Comments for the Author):

The authors used a combination of bioinformatics and functional genomics approaches to study a transcription factor, NagR, as a negative global regulator for pathways involving HMOs utilization in bifidobacteria. The authors reported an altered transcriptional profile in NagR mutant strain and also demonstrated that the phosphorylated derivatives of GlcNAc might be NagR transcriptional effectors. The authors also looked at the occurrence of NagR in a number of bifidobacterial genomes and suggested a potential revolutionary aspect regarding global transcription regulators involved in foraging host-derived glycans in bifidobacteria. Overall, the authors did a comprehensive analysis revealing some of the associated functions of NagR. Please see the following specific comments.

1. The NagR mutant strain was created by insertional mutagenesis. Did the authors test the stability? For single-crossover mutant, it's necessary to use antibiotic to keep the plasmid integrated within the genome, is that what the authors did? It's not clear in the manuscript.
2. In Fig2, Did the authors include an MRS-CS supplemented with glucose control? I assume antibiotic was used in the mutant medium but not in the Wt medium? Did the antibiotic supplementation affect the growth pattern?
3. All the growth curve (y-axis) needs to be in log scale to accurately represent the growth pattern. Throughout the manuscript (e.g. line 144-163), the authors used terms such as "growth rate", "decreased/increased growth" to describe the growth patterns between WT and mutant in MRS-CS supplemented with different carbohydrates. However, the authors did not calculate any growth rates to support this conclusion/observation. I strongly urge the authors to replot the growth pattern in log scale, calculate the growth rate and use that data to describe the growth differences (instead of using individual OD timepoints).
4. For the RNA-seq analysis, I am not sure if an insertional mutant is an ideal candidate for comparative transcriptional analysis with the WT. The RNA-seq results might be skewed due to factors such as antibiotic supplementation in the mutant growth medium and the presence of additional genes (from the plasmid backbones). Did the author have any control over this matter? Perhaps, a potential control could be the same plasmid single crossover into an intergenic region (without disrupting NagR).
5. Line 401, Please briefly describe the transformation protocol.

Reviewer #2 (Comments for the Author):

This manuscript presents a well-written report of the demonstration of the function of NagR proceeding from previous genome analysis of a smaller set of genomes to the current analysis of 25 genomes (including 18 Bifidobacterial species), then to nagR gene knock out, RNA sequencing and testing compounds that interfere with the interaction of NagR with binding sites (operators) by Electrophoretic Mobility Shift Assay.

Overall, a nice addition to understanding the adaptation of Bifidobacteria to specific substrates according to the diet of the host.

Specific comments:

Main text:

Line 283: Citations of references in sentences should not be in parentheses, the authors need to add "as in the previous study (44)".

Line 315: Certain elements of this section (Evolution of the NagR regulon in bifidobacteria) seem to be highly speculative versus supported by the data or the literature. This section should be refined to clarify what has been shown versus what is speculation.

Tables of results should not be cited in the discussion.

Supplementary Methods:

L37: The supplementary methods do not specify how much culture was used to collect cell pellets for RNA isolation (appropriate for the volume of reagents used).

L134-137: Clustering must be accompanied by specific measures of identity from alignments in comparison with the genome similarity in order to advance hypotheses concerning horizontal gene transfer.

July 19, 2022

Dear Dr Lindemann,

Thank you very much for giving us an opportunity to improve our manuscript entitled "**Human Milk Oligosaccharide Utilization in Intestinal Bifidobacteria is Governed by a Global Transcriptional Regulator NagR**". Our responses to all critical comments of the Reviewers are provided below.

Reviewer 1:

Comment 1. *The NagR mutant strain was created by insertional mutagenesis. Did the authors test the stability? For single-crossover mutant, it's necessary to use antibiotic to keep the plasmid integrated within the genome, is that what the authors did? It's not clear in the manuscript.*

Our response: We used a medium with spectinomycin (Sp) only to cultivate overnight cultures of the *nagR*-KO mutant. The mutant was inoculated into an **antibiotic-free MRS-CS medium** for all subsequent experiments (growth curves, metabolic profiling, and RNA-seq). The absence of the antibiotic in these experiments was dictated by the need to compare with a wild-type strain since the presence of Sp in the medium slows down the growth and affects the transcriptome. Excluding Sp from the growth media is essential to delineate the effect of *nagR* inactivation on the *B. infantis* ATCC 15697 transcriptome. The following sentences in the Methods and Supplementary Methods were modified to clarify the usage of Sp in all experiments:

Transcriptome analysis (RNA-seq)

"Overnight cultures of B. infantis ATCC 15697 WT and nagR-KO strains grown in MRS-CS-Lac (with Sp in the case of nagR-KO) were harvested, washed with sugar-free MRS-CS, and used to inoculate antibiotic-free MRS-CS medium supplemented with either Lac or LNnT (1% w/v) at OD₆₀₀=0.02."

Analysis of growth, HMO consumption, and organic acid production

"B. infantis ATCC 15697 WT and nagR-KO strains grown in GAM were harvested, washed twice with sugar-free MRS-CS, and then used to inoculate antibiotic-free MRS-CS medium supplemented with 1% w/v of a specific carbohydrate (Lac, Gal, HMO mixture, sucrose, or FOS) at OD₆₀₀=0.01."

To assess the insertion stability, we have tested the antibiotic susceptibility of the *nagR*-KO mutant after ~30 generations in the antibiotic-free MRS-CS-Lac medium by spreading an aliquot onto Sp⁻ GAM plates, followed by parallel spotting of 100 colonies onto Sp⁺ and Sp⁻ GAM plates. The newly added **Fig. S2C** demonstrates that all 100 colonies grew on Sp⁺ plates, directly confirming insertion stability in the absence of Sp pressure for at least 30 generations, which is more than the number of generations required to grow the cells to the early-log phase as in our RNA-seq experiments. The following sentence was added to the "*Engineered nagR insertional mutant displays comparable yet distinct physiological properties compared to the parental wild-type strain*" section:

"The Sp^R gene insertion at the nagR locus was maintained in the nagR-KO genome for at least 30 generations without antibiotic pressure (Fig. S2C)."

The following paragraph was added to Supplementary methods:

Stability of the Sp^R gene insertion in the absence of the antibiotic pressure

"Overnight culture of the B. infantis ATCC 15697 nagR-KO strain grown in the presence of 15 µg/ml Sp was used to inoculate antibiotic-free MRS-CS-Lac at OD₆₀₀=0.002. Upon reaching OD₆₀₀≈2 (10 estimated generations), the culture was diluted 1000-fold with fresh MRS-CS-Lac"

and grown again to $OD_{600} \approx 2$. The dilution/cultivation cycle was repeated to obtain a culture of approximately 30 generations, and an aliquot was spread onto GAM plates without supplementation of Sp. One hundred colonies that appeared on the plates were then picked up, and their susceptibility to Sp was checked by parallel spotting onto Sp⁻ and Sp⁺ GAM plates."

Comment 2. *In Fig2, Did the authors include an MRS-CS supplemented with glucose control? I assume antibiotic was used in the mutant medium but not in the Wt medium? Did the antibiotic supplementation affect the growth pattern?*

Our response: We omitted the MRC-CS-glucose (Glc) control since the WT *B. infantis* ATCC 15697 strain possesses an insertion mutation in the anti-terminator gene that regulates the glucose-specific PTS transporter and thus has a limited ability to assimilate Glc (PMID: 32985563). In the revised manuscript, we added growth data for WT and *nagR*-KO strains grown on galactose (Gal). Overall, the AUC value (see the next comment) for the mutant is significantly higher than for WT, but the effect size is quite modest (**Fig. 2B**).

As mentioned before, we did not use an antibiotic-supplemented medium for culturing the mutant except for night cultures. Based on our previous experience, Sp supplementation slows down the growth of the mutant.

Comment 3. *All the growth curve (y-axis) needs to be in log scale to accurately represent the growth pattern. Throughout the manuscript (e.g. line 144-163), the authors used terms such as "growth rate", "decreased/increased growth" to describe the growth patterns between WT and mutant in MRS-CS supplemented with different carbohydrates. However, the authors did not calculate any growth rates to support this conclusion/observation. I strongly urge the authors to replot the growth pattern in log scale, calculate the growth rate and use that data to describe the growth differences (instead of using individual OD timepoints).*

Our response: We agree that using the terms "growth rate" and "decreased/increased growth" with respect to our growth data was not well justified. To represent the growth results more rigorously, we have calculated empirical Area Under the Curve (AUC) values by integrating the areas of the trapezoids defined by connecting consecutive data points of absorbance measurements (**Fig. 2B**). AUC is a robust metric of growth integrating the contributions of the initial population size, growth rate in a single value (PMID: 27094401). We have rephrased the sentences where the respective terms were used and replaced. The revised text reads:

*"We used empirical **Area Under the Curve (AUC)** to integrate information from growth curves of *nagR*-KO and wild-type (WT) strains cultivated in the MRS-CS medium supplemented with various carbon sources (**Fig. 2A**). The **AUC values** for *nagR*-KO and WT grown in MRS-CS-Lac or MRS-CS-HMO were not significantly different (**Fig. 2B**). In contrast, the *nagR*-KO mutant had significantly **lower AUC** when grown in MRS-CS supplemented with sucrose and significantly **higher AUC** in the medium containing fructooligosaccharides (FOS) (**Fig. 2B**)."*

Comment 4. *For the RNA-seq analysis, I am not sure if an insertional mutant is an ideal candidate for comparative transcriptional analysis with the WT. The RNA-seq results might be skewed due to factors such as antibiotic supplementation in the mutant growth medium and the presence of additional genes (from the plasmid backbones). Did the author have any control over this matter? Perhaps, a potential control could be the same plasmid single crossover into an intergenic region (without disrupting NagR).*

As already emphasized, to avoid the Sp impact of on the transcriptome of the *nagR*-KO strain we have excluded antibiotic supplementation from our RNA-seq experiments. We agree that the

expression of a few vector-encoded genes could potentially affect the mutant's transcriptome. However, this problem, which is inherent for most available genetic tools, is generally considered to have a very limited impact. Historically, insertional mutants have been widely used to study transcriptional regulators in bifidobacteria, specifically in *Bifidobacterium breve* UCC2003, produced robust and consistent results (as in PMIDs: 24705323, 25688064, 27590817, 29500268).

Comment 5. Line 401, Please briefly describe the transformation protocol.

Our response: We have added the transformation protocol to the "Targeted nagR gene disruption in *B. infantis* ATCC 15697" section:

"To prepare electrocompetent cells, B. infantis ATCC 15697 was grown in 50 mL of Gifu anaerobic medium (GAM; Nissui Pharmaceutical, Tokyo, Japan) to OD₆₀₀=0.2. Cells were harvested by centrifugation (4,800 × g for 15 min at 4 °C), washed with ice-cold 1 mM ammonium citrate buffer (pH 6.0) containing 50 mM sucrose, and resuspended in 400 μL of the same buffer. An aliquot (200 μL) was mixed with 10 μg of the suicide plasmid and then pulsed using a Gene Pulser Xcell system (Bio-Rad Laboratories, Hercules, CA, USA) with 10 kV/cm, 25 μF, and 200 Ω. The pulsed cells were immediately mixed with 800 μL of 1 % (w/v) Lac-supplemented GAM and incubated at 37 °C under anoxic conditions for 3 h before spreading on GAM agar plates containing 1 % (w/v) Lac and 15 μg/mL spectinomycin. Colonies that appeared on the plates were subsequently subjected to a genomic PCR analysis at the nagR locus using primers NagR_III/NagR_IV."

Reviewer 2:

Comment 1. Line 283: Citations of references in sentences should not be in parentheses, the authors need to add "as in the previous study (44)".

Our response: We thank the Reviewer for the comment. We have added the suggested sentence.

Comment 2. Line 315: Certain elements of this section (Evolution of the NagR regulon in bifidobacteria) seem to be highly speculative versus supported by the data or the literature. This section should be refined to clarify what has been shown versus what is speculation.

Our response: We appreciate and agree with this criticism. We have rewritten the respective section to distinguish conclusions about the evolution of regulatory networks supported by the data (making a distinction between bioinformatic predictions and experimental data) and our speculative conjectures.

Comment 3. Tables of results should not be cited in the discussion.

Our response: We have decided to delete Table 2 since it mostly repeats information in the discussion section. The links to this table in the discussion have been deleted.

Comment 4. L37: The supplementary methods do not specify how much culture was used to collect cell pellets for RNA isolation (appropriate for the volume of reagents used).

Our response: We have clarified the quantity of a pelleted cell culture used for RNA isolation. The text now reads:

"Frozen cell pellets obtained from 2 ml of early-mid exponential phase cultures (OD₆₀₀=0.35) were kept on dry ice and resuspended in a solution containing 250 μL of acid-washed glass beads (212-300 μm), 710 μL of extraction mixture (200 mM NaCl, 20 mM EDTA, 20% SDS), and 500 μL of a mixture of phenol:chloroform:isoamyl alcohol (125:24:1, pH 4.5)."

Comment 5. L134-137: Clustering must be accompanied by specific measures of identity from alignments in comparison with the genome similarity in order to advance hypotheses concerning horizontal gene transfer.

Our response: We have modified the text to include an alternative hypothesis of convergent evolution. However, we still think the potential horizontal transfer is more likely given the conservation of the gene order in *BITS_RS02005-nagR* < *gltABC-lnpABC* divergon in both species. The conservation of gene order between distant taxa strongly indicates HGT (PMID: 11544372). The revised text reads as follows:

"We also noted a discrepancy between positions of branches corresponding to A. theropithecii in species (Fig. 6A) and gene (Fig. 6B) phylogenetic trees. The A. theropithecii branch was an outgroup in the species tree, reflecting the phylogenetic differences between Bifidobacterium and Alloscardovia genera. However, in the gene tree, NagR from A. theropithecii clustered with NagR sequences from bifidobacteria isolated from mammals (e.g., Bifidobacterium tsurumiense JCM 13495). Notably, the genomic neighborhood of the nagR genes was also surprisingly similar in both species: a BITS_RS02005-nagR operon forming a divergon with gltABC-lnpABC (Table S3). This observation may be indicative of a possible horizontal transfer of the whole gene cluster from Bifidobacterium spp. to A. theropithecii or convergent evolution in a mammalian host."

Sincerely,

Aleksandr Arzamasov and Andrei Osterman, Sanford Burnham Prebys Medical Discovery Institute (with contribution and approval of all other co-authors).

August 23, 2022

Dr. Andrei L. Osterman
Sanford Burnham Prebys Medical Discovery Institute
La Jolla

Re: mSystems00343-22R1 (Human Milk Oligosaccharide Utilization in Intestinal Bifidobacteria is Governed by a Global Transcriptional Regulator NagR)

Dear Dr. Andrei L. Osterman:

Your manuscript has been accepted, and I am forwarding it to the ASM Journals Department for publication. For your reference, ASM Journals' address is given below. Before it can be scheduled for publication, your manuscript will be checked by the mSystems production staff to make sure that all elements meet the technical requirements for publication. They will contact you if anything needs to be revised before copyediting and production can begin. Otherwise, you will be notified when your proofs are ready to be viewed.

Publication Fees:

If you would like to submit a potential Featured Image, please email a file and a short legend to mSystems@asmusa.org. Please note that we can only consider images that (i) the authors created or own and (ii) have not been previously published. By submitting, you agree that the image can be used under the same terms as the published article. File requirements: square dimensions (4" x 4"), 300 dpi resolution, RGB colorspace, TIF file format.

We recognize that the video files can become quite large, and so to avoid quality loss ASM suggests sending the video file via <https://www.wetransfer.com/>. When you have a final version of the video and the still ready to share, please send it to mSystems staff at mSystems@asmusa.org.

Sincerely,

Stephen Lindemann
Editor, mSystems

Journals Department
Table S1: Accept
Fig. S2: Accept
Supplementary Text: Accept
Fig. S1: Accept
Table S3: Accept
Fig. S4: Accept
Fig. S3: Accept
Table S2: Accept
Fig. S5: Accept
Supplementary Code File: Accept